# PRIVACY-PRESERVING DATA QUALITY EVALUATION IN FEDERATED LEARNING USING INFLUENCE APPROXIMATION

## ABSTRACT

In Federated Learning, it is crucial to handle low-quality, corrupted, or malicious data, but traditional data valuation methods are not suitable due to privacy concerns. To address this, we propose a simple yet effective approach that utilizes a new influence approximation called "lazy influence" to filter and score data while preserving privacy. To do this, each participant uses their own data to estimate the influence of another participant's batch and sends a differentially private obfuscated score to the Center of the federation. Our method has been shown to successfully filter out corrupted data in various applications, achieving a recall rate of over $> 90\%$ (sometimes up to $100\%$) while maintaining strong differential privacy guarantees with epsilon values of less than or equal to one.

## 1 INTRODUCTION

The success of Machine Learning (ML) depends to a large extent on the availability of high-quality data. This is a critical issue in Federated Learning (FL) since the model is trained without access to raw training data. Instead, a single *Center* uses data from independent and sometimes self-interested *data holders* to train a model jointly. Having the ability to *score* and *filter* irrelevant, noisy, or malicious data can (i) significantly improve model accuracy, (ii) speed up training, and even (iii) reduce costs for the Center when it pays for data.

Federated Learning McMahan et al. (2017a); Kairouz et al. (2021b); Wang et al. (2021) differs from traditional centralized ML approaches. Challenges such as scalability, communication efficiency, and privacy can no longer be treated as an afterthought; instead, they are *inherent constraints* of the setting. For example, data holders often operate resource-constrained edge devices and include businesses and medical institutions that must protect the privacy of their data due to confidentiality or legal constraints.

A clean way of quantifying the effect of data point(s) on the accuracy of a model is via the notion of *influence* Koh & Liang (2017); Cook & Weisberg (1980). Intuitively, influence quantifies the marginal contribution of a data point (or batch of points) on a model's accuracy. One can compute this by comparing the difference in the model's empirical risk when trained with and without the point in question. While the influence metric can be highly informative, it is impractical to compute: re-training a model is time-consuming, costly, and often impossible, as participants do not have access to the entire dataset. We propose a simple and practical approximation of the *sign* of the exact influence (*lazy influence*), which is based on an estimate of the direction of the model after a small number of local training epochs with the new data.

Another challenge is to approximate the influence while preserving the privacy of the data. Many approaches to Federated Learning (e.g., McMahan et al. (2018); Triastcyn & Faltings (2019)) remedy this by combining FL with Differential Privacy (DP) Dwork (2006b;a); Dwork et al. (2006a;b); Kairouz et al. (2021a); De et al. (2022); Choquette-Choo et al. (2022), a data anonymization technique that many researchers view as the gold standard Triastcyn (2020). We show how the sign of influence can be approximated in an FL setting while maintaining strong differential privacy guarantees. Specifically, there are two sets of participants' data that we need to protect: the training and the validation data (see also Section 1.2). We clip and add noise to the gradients for the evaluated training data according to McMahan et al. (2017b), which achieves a *local* differential privacy guar-

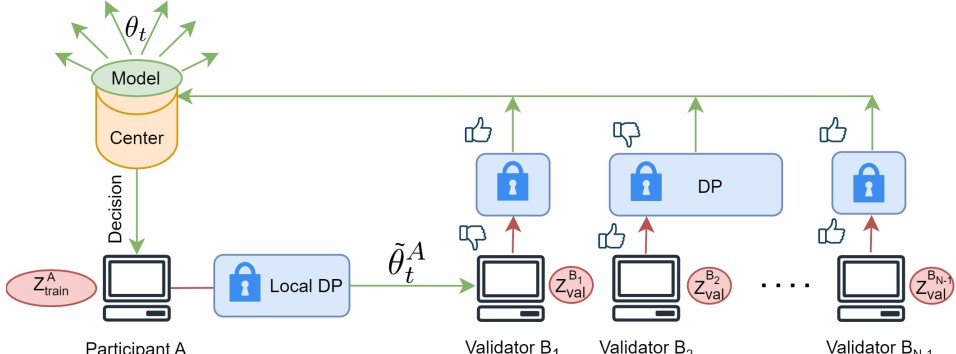

Figure 1: Data filtering procedure. A Center heads a federation of participants $A, B_1, .., B_{N-1}$ that holds private data relevant to the joint model. $A$ sends an obfuscated *lazy* (i.e., partial/approximate) model update to $B_i$, who submit a vote based on their own validation data. The differentially private aggregated votes are used to decide whether to incorporate $A$'s data. See Section 1.2.

antee. To ensure the privacy of the validation data and the influence approximation itself, we employ a differentially private defense mechanism based on the idea of randomized response Warner (1965) (inspired by Erlingsson et al. (2014)). Together the two mechanisms ensure strong, *worst-case privacy* guarantees while allowing for accurate data filtering.

The proposed approach can be used as a 'right of passage' every time a participant joins the federation, periodically during communication rounds (most resource intensive, but would provide the best results), or even as a diagnostic tool. A quality score is helpful for various purposes beyond filtering poor data, such as rewarding the data provider, incentivizing users in a crowdsourcing application, assessing a data provider's reputation, etc.

## 1.1 Our Contributions

We address two major challenges in this work: (i) efficiently estimating the quality of a batch of training data and (ii) keeping both the training and validation data used for this estimate private. For the former, we develop a novel metric called *lazy influence*, while for the latter, we add noise to the gradients and propose a differentially private voting scheme. More specifically:

**(1)** We present a novel technique (*lazy influence approximation*) for scoring and filtering data in Federated Learning.

**(2)** We show that our proposed distributed influence aggregation scheme allows for robust scoring, even under rigorous, worst-case differential privacy guarantees (privacy cost $\varepsilon < 1$). This is the recommended value in DP literature and much smaller than many other AI or ML applications.[1]

**(3)** We evaluate our approach on two well-established datasets (CIFAR10 and CIFAR100) and demonstrate that **filtering using our scheme can eliminate the adverse effects of inaccurate data**.

## 1.2 High-Level Description of Our Setting

A center $C$ coordinates a set of participants to train a single model (Figure 1). $C$ has a small set of 'warm-up' data, which are used to train an initial model $M_0$ that captures the desired input/output relation. We assume that each data holder has a set of training points that will be used to improve the model and a set of validation points that will be used to evaluate other participants' contributions. It must be kept private to prohibit participants from tailoring their contributions to the validation data. For each federated learning round $t$ (model $M_t$), each data holder participant will assume two roles: the role of the contributor ($A$) and the role of the validator ($B$). As a contributor, a participant

---

[1]AI or ML applications often assume $\varepsilon$ as large as 10 Triastcyn & Faltings (2019) (see, e.g., Tang et al. (2017)). For specific attacks, $\varepsilon = 10$ means that an adversary can theoretically reach an accuracy of 99.99% Triastcyn & Faltings (2019)

performs a small number of local epochs to $M_t$ – enough to get an estimate of the gradient[2] – using a batch of his training data $z_{A,t}$. Subsequently, $A$ sends the updated partial model $M_{t,A}$, with specifically crafted noise to ensure local DP, to every other participant (which assumes the role of a validator). The applied noise protects the updated gradient while still retaining information on the usefulness of data. Each validator $B$ uses its validation dataset to approximate the empirical risk of $A$'s training batch (i.e., the approximate influence). This is done by evaluating each validation point and comparing the loss. In an FL setting, we can not re-train the model to compute the exact influence; instead, $B$ performs only a small number of training epochs, enough to estimate the direction of the model (lazy influence approximation). As such, we look at the sign of the approximate influence (and not the magnitude). Each validator aggregates the signs of the influence for each validation point, applies controlled noise to ensure DP, and sends this information to the center. Finally, the center accepts $A$'s training batch if most of the $B$s report positive influence and reject otherwise.

## 2   RELATED WORK AND DISCUSSION

**Federated Learning**   Federated Learning (FL) McMahan et al. (2017a); Kairouz et al. (2021b); Wang et al. (2021); Li et al. (2020) has emerged as an alternative method to train ML models on data obtained by many different agents. In FL, a center coordinates agents who acquire data and provide model updates. FL has been receiving increasing attention in both academia Lim et al. (2020); Yang et al. (2019); He et al. (2020); Caldas et al. (2018) and industry Hard et al. (2018); Chen et al. (2019), with a plethora of real-world applications (e.g., training models from smartphone data, IoT devices, sensors, etc.).

**Influence functions**   Influence functions are a standard method from robust statistics Cook & Weisberg (1980) (see also Section 3), which were recently used as a method of explaining the predictions of black-box models Koh & Liang (2017). They have also been used in the context of fast cross-validation in kernel methods and model robustness Liu et al. (2014); Christmann & Steinwart (2004). While a powerful tool, computing the influence involves too much computation and communication, and it requires access to the training and validation data (see Koh & Liang (2017) and Section 3). There has also been recent work trying to combine Federated Learning with influence functions Xue et al. (2021), though to the best of our knowledge, we are the first to provide a privacy-preserving alternative.

**Data Filtering**   A common but computationally expensive approach for filtering in ML is to use the Shapley Value of the Influence to evaluate the quality of data Jia et al. (2019b); Ghorbani & Zou (2019a); Jia et al. (2019a); Yan et al. (2020); Ghorbani & Zou (2019b); Watson et al. (2022). Other work includes, for example, rule-based filtering of least influential points Ogawa et al. (2013), or constructing weighted data subsets (corsets) Dasgupta et al. (2009). Because of the privacy requirements in FL, contributed data is not directly accessible for assessing its quality. Tuor et al. (2021) propose a decentralized filtering process specific to federated learning, yet they do not provide any formal privacy guarantees.

While data filtering might not always pose a significant problem in traditional ML, in an FL setting, it is more important because even a small percentage of mislabeled data can result in a significant drop in the combined model's accuracy.

**Client Selection and Attack Detection**   Our setting can also be interpreted as potentially adversarial, but it should not be confused with Byzantine robustness. We do not consider threat scenarios as described in Cao et al. (2020) and So et al. (2020), where participants carefully craft malicious updates. Instead, we assume that the data used for those updates might be corrupt. For completeness and in lack of more relevant baselines, we compare our work to two Byzantine robust methods: KRUM Blanchard et al. (2017), Trimmed-mean Yin et al. (2018), and Centered-Clipping Karimireddy et al. (2021) (along to an oracle filter). These methods, though, require gradients to be transmitted as is, i.e., lacking any formal privacy guarantee to the participants' training data. Furthermore, both of these techniques require the center to know the number of malicious participants

---

[2]The number of local epochs is a hyperparameter. We do not need to train the model fully. See Section 3.2.

a priori. Another important drawback is that they completely eliminate "minority" distributions due to their large distance relative to other model updates.

**Differential Privacy**   Differential Privacy (DP) Dwork (2006b;a); Dwork et al. (2006a;b) has emerged as the de facto standard for protecting the privacy of individuals. Informally, DP captures the increased risk to an individual's privacy incurred by participating in the learning process. Consider a participant being surveyed on a sensitive topic as a simplified, intuitive example. To achieve differential privacy, one needs a source of randomness; thus, the participant decides to flip a coin. Depending on the result (heads or tails), the participant can reply truthfully or randomly. An attacker can not know if the decision was taken based on the participant's preference or due to the coin toss. Of course, to get meaningful results, we need to bias the coin toward the actual data. In this simple example, the logarithm of the ratio $Pr[\text{heads}]/Pr[\text{tails}]$ represents the privacy cost (also referred to as the privacy budget), denoted traditionally by $\varepsilon$. Yet, one must be careful in designing a DP mechanism, as it is often hard to practically achieve a meaningful privacy guarantee (i.e., avoid adding a lot of noise and maintain high accuracy) Triastcyn & Faltings (2019); Danassis et al. (2022). A variation of DP, instrumental in our context, given the decentralized nature of federated learning, is Local Differential Privacy (LDP) Dwork et al. (2014). LDP is a generalization of DP that provides a bound on the outcome probabilities for any pair of individual participants rather than populations differing on a single participant. Intuitively, it means that one cannot hide in the crowd. Another strength of LDP is that it does not use a centralized model to add noise–participants sanitize their data themselves– providing privacy protection against a malicious data curator. For a more comprehensive overview of DP, we refer the reader to Triastcyn (2020); Dwork et al. (2014). We assume that the participants and the Center are *honest but curious*, i.e., they don't actively attempt to corrupt the protocol but will try to learn about each other's data.

## 3   METHODOLOGY

We aim to address two challenges: (i) approximating the influence of a (batch of) data point(s) without having to re-train the entire model from scratch and (ii) doing so while protecting the privacy of training and validation data. The latter is essential not only to protect users' sensitive information but also to ensure that malicious participants can not tailor their contributions to the validation data. We first introduce the notion of *influence* Cook & Weisberg (1980) (for detailed definitions please see the supplement) and our proposed lazy approximation. Second, we describe a differentially private reporting scheme for crowdsourcing the approximate influence values.

**Setting**   We consider a classification problem from some input space $\mathcal{X}$ (e.g., features, images, etc.) to an output space $\mathcal{Y}$ (e.g., labels). In a FL setting, there is a center $C$ that wants to learn a model $M(\theta)$ parameterized by $\theta \in \Theta$, with a non-negative loss function $L(z, \theta)$ on a sample $z = (\bar{x}, y) \in \mathcal{X} \times \mathcal{Y}$. Let $R(Z, \theta) = \frac{1}{n}\sum_{i=1}^{n} L(z_i, \theta)$ denote the empirical risk, given a set of data $Z = \{z_i\}_{i=1}^{n}$. We assume that the empirical risk is differentiable in $\theta$. The training data are supplied by a set of data holders.

### 3.1   Shortcomings of the Exact and Approximate Influence in a FL Setting

**Definitions**   In simple terms, influence measures the marginal contribution of a data point on a model's accuracy. A positive influence value indicates that a data point improves model accuracy, and vice-versa. More specifically, let $Z = \{z_i\}_{i=1}^{n}$, $Z_{+j} = Z \cup z_j$ where $z_j \notin Z$, and let $\hat{R} = \min_\theta R(Z, \theta)$ and $\hat{R}_{+j} = \min_\theta R(Z_{+j}, \theta)$, where $\hat{R}$ and $\hat{R}_{+j}$ denote the minimum empirical risk of their respective set of data. The *influence* of datapoint $z_j$ on $Z$ is defined as $\mathcal{I}(z_j, Z) \triangleq \hat{R} - \hat{R}_{+j}$

Despite being highly informative, influence functions have not achieved widespread use in FL (or ML in general). This is mainly due to the computational cost. The exact influence requires complete retraining of the model, which is time-consuming and very costly, especially for state-of-the-art, large ML models (specifically for our setting, we do not have direct access to the training data). Recently, the first-order Taylor approximation of influence Koh & Liang (2017) (based on Cook & Weisberg (1982)) has been proposed as a practical method to understanding the effects of training points on the predictions of a *centralized* ML model. While it can be computed without having to re-train the model, according to the following equation $\mathcal{I}_{appr}(z_j, z_{val}) \triangleq$

---

**Algorithm 1:** Filtering Poor Data Using Lazy Influence Approximation in Federated Learning

---

**Data:** $\theta_0$, $Z_i$, $Z_{val}$, $Z_{init}$
**Result:** $\theta_T$

**1** $C$: The center ($C$) initializes the model $M_0(\theta_0)$
**2** **for** $t \in T$ *rounds of Federated Learning* **do**
**3**     $C$: Broadcasts $\theta_t$
**4**     **for** $P_i$ *in* $Participants$ **do**
**5**        $P_i$: Acts as a contributor ($A$). Performs $k$ local epochs with $Z_{A,t}$ on the partially-frozen
          model $\tilde{\theta}_t^A$.
**6**        $P_i$: Applies DP noise to $\tilde{\theta}_t^A$.
**7**        $P_i$: sends last layer of $\tilde{\theta}_t^A$ to $Participants_{-i}$.
**8**        **for** $P_j$ *in* $Participants_{-i}$ **do**
**9**           $P_j$: Acts as a validator ($B$). Evaluates the loss of $Z_{val}^B$ on $\theta_t$
**10**          $P_j$: Evaluates the loss of $Z_{val}^B$ on $\tilde{\theta}_t^A$
**11**          $P_j$: Calculates vote $v$ (sign of influence), according to Equation 1
**12**          $P_j$: Applies noise to $v$ according to his privacy parameter $p$ to get $v'$ (Equation 2)
**13**          $P_j$: Sends $v'$ to $C$
**14**        $C$: Filters out $P_i$'s data based on the votes from $Participants_{-i}$ (i.e., if
         $\sum_{\forall B} I_{proposed}(Z_{val}^B) < T$).
**15**     $C$: Updates $\theta_t$ using data from unfiltered $Participants$;

---

$-\nabla_\theta L(z_{val}, \hat{\theta}) H_{\hat{\theta}}^{-1} \nabla_\theta L(z_j, \hat{\theta})$, it is still ill-matched for FL models for several key reasons, as explained in the following paragraph.

**Challenges** To begin with, computing the influence approximation of Koh & Liang (2017) requires *forming and inverting* the Hessian of the empirical risk. With $n$ training points and $\theta \in \mathbb{R}^m$, this requires $O(nm^2 + m^3)$ operations Koh & Liang (2017), which is *impractical* for modern-day deep neural networks with millions of parameters. To overcome these challenges, Koh & Liang (2017) used implicit Hessian-vector products (HVPs) to more efficiently approximate $\nabla_\theta L(z_{val}, \hat{\theta}) H_{\hat{\theta}}^{-1}$, which typically requires $O(p)$ Koh & Liang (2017). While this is a somewhat more efficient computation, it is *communication-intensive*, as it requires *transferring all of the (either training or validation) data* at each FL round. Most importantly, it *can not provide any privacy* to the users' data, an important, inherent requirement/constraint in FL. Finally, the loss function has to be strictly convex and twice differentiable (which is not always the case in modern ML applications). The proposed solution is to swap out non-differentiable components for smoothed approximations, but there is no quality guarantee of the influence calculated in this way.

### 3.2 LAZY INFLUENCE: A PRACTICAL INFLUENCE METRIC FOR FILTERING DATA IN FL APPLICATIONS

The key idea is that *we do not need to approximate the influence value* to filter data; we only need an accurate estimate of its *sign* (in expectation). Recall that a positive influence value indicates a data point improves model accuracy. Thus, we only need to approximate the sign of loss and use that information to filter out data whose influence falls below a certain threshold.

Recall that each data holder participant assumes two roles: the role of the contributor ($A$) and the role of the validator ($B$). Our proposed approach works as follows (Algorithm 1):

**(i)** For each federated learning round $t$ (model $M_t(\theta_t)$), the contributor participant $A$ performs a small number $k$ of local epochs to $M_t$ using a batch of his training data $Z_{A,t}$, resulting in $\tilde{\theta}_t^A$. $k$ is a hyperparameter. $\tilde{\theta}_t^A$ is the partially trained model of participant $A$, where most of the layers, except the last one, have been frozen. The model should not be fully trained for two key reasons: efficiency and avoiding over-fitting (e.g., in our simulations, we only performed 1-9 epochs). Furthermore, $A$

adds noise to $\tilde{\theta}_t^A$ (see Section 3.2.2) to ensure strong, worst-case local differential privacy. Finally, $A$ sends only the last layer (to reduce communication cost) of $\tilde{\theta}_t^A$ to every other participant.

**(ii)** Each validator $B$ uses his validation dataset $Z_{val}^B$ to estimate the sign of the influence using Equation 1. Next, the validator applies noise to $I_{proposed}(Z_{val}^B)$, as described in Section 3.2.3, to ensure strong, worst-case differential privacy guarantees (i.e., keep his validation dataset private).

$$\mathcal{I}_{proposed}(Z_{val}^B) \triangleq \text{sign}\left(\sum_{z_{val} \in Z_{val}^B} L(z_{val}, \theta_t) - L(z_{val}, \theta_t^A)\right) \tag{1}$$

**(iii)** Finally, the center $C$ aggregates the obfuscated votes $I_{proposed}(Z_{val}^B)$ from all validators and filters out data with cumulative score *below a threshold* ($\sum_{\forall B} I_{proposed}(Z_{val}^B) < T$). Specifically, we cluster the votes into two clusters (using k-means) and use the arithmetic mean of the cluster centers as the filtration threshold.

### 3.2.1 Advantages of the proposed lazy influence

Depending on the application, the designer may select any optimizer to perform the model updates. We do not require the loss function to be twice differentiable and convex; only once differentiable. It is significantly more *computation and communication efficient*; an essential prerequisite for any FL application. This is because participant $A$ only needs to send (a *small part* of) the model parameters $\theta$, not his training data. Moreover, computing a few model updates (using, e.g., SGD or any other optimizer) is significantly faster than computing either the exact influence or an approximation due to the numerous challenges (please refer to the supplementary materials for a detailed description). Finally, and importantly, we ensure the *privacy* of both the train and validation dataset of every participant.

### 3.2.2 Sharing the Partially Updated Joint Model: Privacy and Communication Cost

Each contributor $A$ shares a partially trained model $\tilde{\theta}_t^A$ (see step (i) of Section 3.2). It is important to stress that A only sends the last layer of the model. This has two significant benefits: it *reduces the communication overhead* (in our simulations, *we only send $0.009\%$ of the model's weights*),[3] and minimize the impact of the differential privacy noise. We follow McMahan et al. (2017b) to ensure strong local differential privacy guarantees by (i) imposing a bound on the gradient (using a clipping threshold $\Delta$), and (ii) adding carefully crafted Gaussian noise (parameterized by $\sigma$). For more details, see McMahan et al. (2017b).

### 3.2.3 Differentially Private Reporting of the Influence

Along with the training data, we also need to ensure the privacy of the validation data used to calculate the influence. Protecting the validation data in an FL setting is critical since (i) it is an important constraint of the FL setting, (ii) participants want to keep their sensitive information (and potential means of income, e.g., in a crowdsourcing application) private, and (iii) the center wants to ensure that malicious participants can not tailor their contributions to the validation set.

We obfuscate the influence reports using RAPPOR Erlingsson et al. (2014), which results in an $\varepsilon$-differential privacy guarantee Dwork et al. (2006b). The obfuscation (permanent randomized response Warner (1965)) takes as input the participant's true influence value $v$ (binary) and privacy parameter $p$, and creates an obfuscated (noisy) reporting value $v'$, according to Equation 2. $p$ is a *user-tunable* parameter that allows the participants themselves to *choose their desired level of privacy*, while maintaining reliable filtering. The worst-case privacy guarantee can be computed by each participant *a priori*, using Equation 3 Erlingsson et al. (2014).

$$v' = \begin{cases} +1, & \text{with probability } \frac{1}{2}p \\ -1, & \text{with probability } \frac{1}{2}p \\ v, & \text{with probability } 1-p \end{cases} \tag{2} \qquad\qquad \varepsilon = 2\ln\left(\frac{1 - \frac{1}{2}p}{\frac{1}{2}p}\right) \tag{3}$$

---

[3]Moreover, as explained in the Introduction, this communication cost will be incurred as little as *one* time, when we use our approach as a 'right of passage' every time a participant joins the federation.

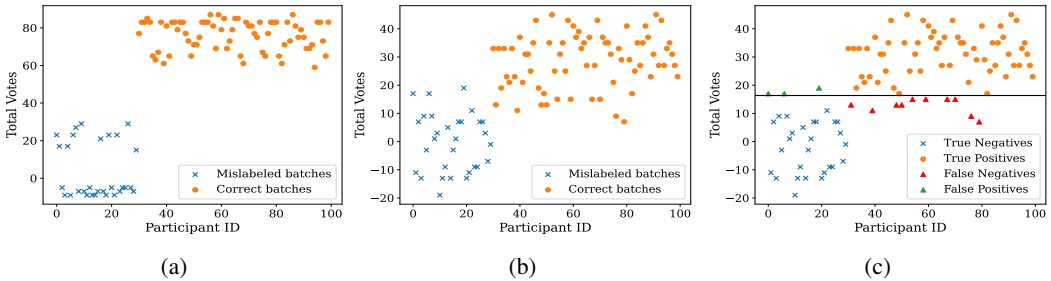

Figure 2: Visualization of the voting scheme. The $x$-axis represents a contributor participant $A$. The $y$-axis shows the sum of all votes from all the validators, i.e., $\sum_{\forall B} I_{proposed}(Z_{val}^B)$. Figure 2a corresponds to the sum of true votes (no privacy) for the valiation data of each contributor on the $x$-axis, while Figure 2b depicts the sum of differentially private votes ($\epsilon = 1$), according to randomized reporting algorithm (see the supplementary materials for a detailed description). Finally, Figure 2c shows the filtration threshold, corresponding to the arithmetic mean of the two cluster centers (computed using k-means).

It is important to note that in a Federated Learning application, the center $C$ aggregates the influence sign from *a large number of participants*. This means that even under *really strict* privacy guarantees, *the aggregated influence signs (which is exactly what we use for filtering) will match the true value* in expectation. This results in *high-quality filtering*, as we will demonstrate in Section 4.

To demonstrate the effect of Equation 2, we visualize the obfuscation process in Figure 2. Figure 2a shows the sum of true votes ($y$-axis) for the validation data of each contributor ($x$-axis). Here we can see a clear distinction in votes between corrupted and correct batches. Most of the corrupted batches (corrupted contributor participants) take negative values, meaning that the majority of the validators voted against them. In contrast, the correct batches are close to the upper bound. Figure 2b demonstrates the effect of applying DP noise ($\epsilon = 1$) to the votes: differentiating between the two groups becomes more challenging. To find an effective decision threshold, we use k-means to cluster the votes into two clusters and use the arithmetic mean of the cluster centers as the filtration threshold (Figure 2c).

## 4 EVALUATION RESULTS

We evaluated the proposed approach on two well-established datasets: **CIFAR10** Krizhevsky et al. (2009), and **CIFAR100** Krizhevsky et al. (2009). Furthermore, we consider two corruption methods:

1. **Random label**: A random label is sampled for every training point. Used for the IID setting (as it does not make sense to assign a random label to a highly skewed Non-IID setting).

2. **Label shift**: Every correct label is mapped to a different label and this new mapping is applied to the whole training dataset. Used in both IID and non-IID settings.

**Setup** Our evaluation involves a single round of Federated Learning. A small portion of every dataset (around 1%) is selected as the 'warm-up' data used by the center $C$ to train the initial model $M_0$. Each participant has two datasets: a training batch ($Z_A$, see Section 3.2, step (i)), which the participant uses to update the model when acting as the contributor participant, and a validation dataset ($Z_{val}^B$, see Section 3.2, step (ii)), which the participant uses to estimate the sign of the influence when acting as a validator participant. The ratio of these datasets is $2 : 1$. The training batch size is 100 (i.e., the training dataset includes 100 points, and the validation dataset consists of 50 points). This means that, e.g., for a simulation with 100 participants, each training batch is evaluated on $50 \times (100 - 1)$ validation points, and that for each training batch (contributor participant $A$), the center collected $(100 - 1)$ estimates of the influence sign (Equation 1). We corrupted 30% of the total batches (i.e., participants). For each corrupted batch, we corrupted 90% of the data points. Each simulation was run 8 times. We report average values and standard deviations. Please see the supplement for detailed results.

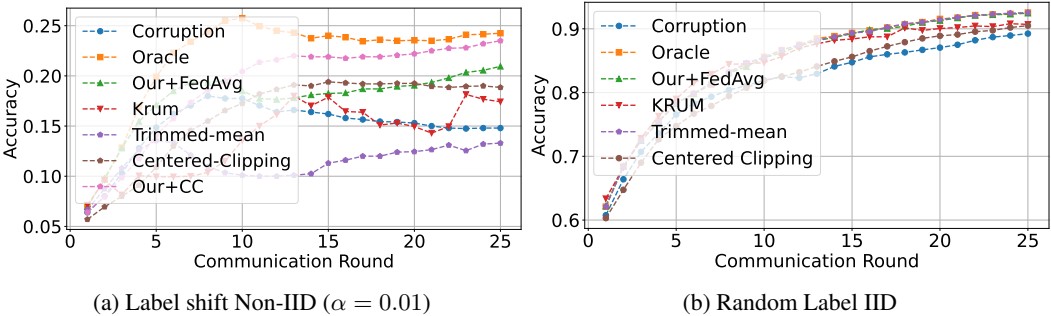

Figure 3: Model accuracy over 25 communication rounds with a 30% mislabel rate on CIFAR-10. We compare a centralized model with no filtering (blue) to an FL model under perfect (oracle) filtering (orange), KRUM (red), Trimmed-mean (purple), Centered-Clipping(brown), our approach with FedAvg (green), and our approach with Centered-Clipping (pink). Note that the jagged line for KRUM is because only a single gradient is selected instead of performing FedAvg.

**Implementation** The proposed approach is model-agnostic and can be used with *any* gradient-descent-based machine learning method. For our simulations, we used a Vision Transformer (ViT), as it exhibits state-of-the-art performance Dosovitskiy et al. (2020) (specifically, HuggingFace's implementation Wolf et al. (2020)).

**Non-IID Setting** The main hurdle for FL is that not all data is IID. Heterogeneous data distributions are all but uncommon in the real world. To simulate non-IID data, we used the Dirichlet distribution to split the training dataset as in related literature Hsu et al. (2019); Lin et al. (2020); Hoech et al. (2022); Yu et al. (2022). This distribution is parameterized by $\alpha$, which controls the concentration of different classes. See the supplement for a visualization. In this work, we use $\alpha \to 0.1$ for a non-IID distribution, as in related literature (e.g., Yu et al. (2022)).

**Baselines** We compare against four baselines: **(i) Corrupted model**: this shows us the training performance of a model which any technique has not sanitized. **(ii) Oracle filtration**: this represents the ideal scenario where we know which participants contribute bad data. **(iii) KRUM**: byzantine robustness technique Blanchard et al. (2017) that selects the best gradient out of the update based on a pair-wise distance metric. **(iv) Trimmed-mean**: another byzantine robustness technique Yin et al. (2018) that takes the average of gradients instead of just selecting one, also based on a pair-wise distance metric (see also Section 2). **(v) Centered-Clipping**: current state-of-the-art technique for byzantine robust aggregatorsKarimireddy et al. (2021). Furthermore, we show that our filtration technique is not mutually exclusive with these aggregators and that it is highly beneficial in addition to them, as can be seen in Figure 3.

## 4.1 Model Accuracy

The proposed approach achieves *high model accuracy, close to the perfect (oracle) filtering* (13.6% worse in the non-IID setting, and 0.1% in the IID setting). Focusing on the non-IID setting (Figure 3a), which is the more challenging and relevant for FL, our approach achieves a 20.3% *improvement over KRUM*, and a 57.5% *improvement over the Trimmed-mean baseline*, after 25 communication rounds. We can also observe that *Centered-Clipping* outperforms all other byzantine robust aggregators. Additionally, combining it with our approach provides results almost as good as the *Oracle filter*. Finally, in the IID setting, all methods perform similarly (Figure 3b), though recall that the baselines do not provide privacy guarantees (see Section 2).

## 4.2 Recall, and Precision of Filtration

Recall is the most informative metric to evaluate the efficiency of our filtering approach. Recall refers to the ratio of detected mislabeled batches over all of the mislabeled batches. *Including a mislabeled batch can harm a model's performance significantly more compared to removing an unaltered batch.* Thus, achieving *high recall* is of paramount importance. Meanwhile, precision

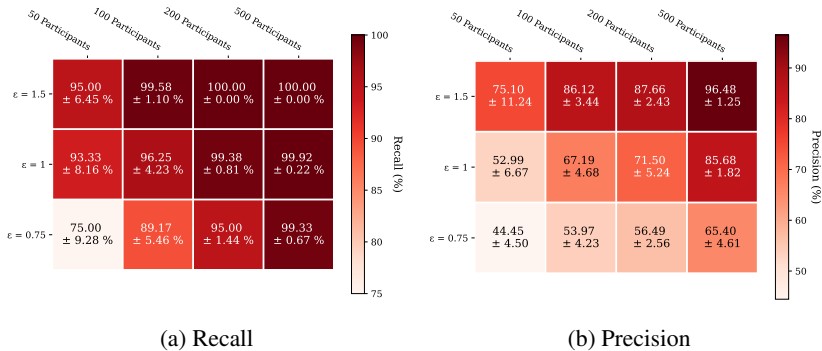

(a) Recall            (b) Precision

Figure 4: Recall (left), and Precision (right) on CIFAR 10, non-IID, for increasing problem size (number of participants), and varying privacy guarantees ($\varepsilon$ – lower $\varepsilon$ provides stronger privacy).

represents the ratio of correctly identified mislabeled batches over all batches identified as mislabeled. An additional benefit of using the proposed lazy influence metric for scoring data is that it also allows us to identify correctly labeled data, which nevertheless do not provide a significant contribution to the model.

The proposed approach achieves both high recall and precision (Figure 4), despite the *high degree of non-IID* (low concentration of classes per participant). Notably, the metrics improve significantly as we increase the number of participants (horizontal axis). In simple terms, more validators mean more samples of the different distributions. Thus, 'honest' participants get over the filtering threshold, even in highly non-IID settings. Recall reaches $100\%$, and precision $96.48\%$ by increasing the number of participants to just 500, in the non-IID setting and under really strict worst-case privacy guarantees. Results for the IID setting are significantly better (please see the supplement).

### 4.3 PRIVACY

As expected, there is a trade-off between privacy and filtration quality (see Figure 4, vertical axis, where $\varepsilon$ refers to the privacy guarantee for both the training and validation data/participant votes). Nevertheless, Figure 4 demonstrates that our approach can provide *reliable filtration*, even under *really strict, worst-case privacy requirements* ($\varepsilon = 1$, which is the recommended value in the DP literature Triastcyn (2020)). Importantly, our decentralized framework allows each participant to *compute* and *tune* his *own* worst-case privacy guarantee *a priori* (see Section 3.2.3).

The *privacy trade-off can be mitigated*, and the quality of the filtration can be significantly improved by increasing the number of participants (Figure 4, horizontal axis). The higher the number of participants, the better the filtration (given a fixed number of corrupted participants). This is because as the number of participants increases, the aggregated influence signs (precisely what we use for filtering) will match the actual value in expectation. For 500 participants, we achieve high-quality filtration even for $\varepsilon = 0.75$. This is important given that in most real-world FL applications, we *expect a large number of participants*.

### 5 CONCLUSION

Privacy protection is a core element of Federated Learning. However, this privacy also means that it is significantly more difficult to ensure that the training data actually improves the model. Mislabeled, corrupted, or even malicious data can result in a strong degradation of the performance of the model – as we also demonstrated empirically – and privacy protection makes it significantly more challenging to identify the cause. In this work, we propose the *"lazy influence"*, a *practical* influence approximation that characterizes the quality of training data and allows for effective filtering (recall of $> 90\%$, and even up to 100% as we increase the number of participants), while providing *strict, worst-case $\varepsilon$-differential privacy guarantees* ($\varepsilon < 1$) for both the training and validation data. The proposed approach can be used to filter bad data, recognize good and bad data providers, and pay data holders according to the quality of their contributions.

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

## A  SETTING

We consider a classification problem from some input space $\mathcal{X}$ (e.g., features, images, etc.) to an output space $\mathcal{Y}$ (e.g., labels). In a Federated Learning setting, there is a center $C$ that wants to learn a model $M(\theta)$ parameterized by $\theta \in \Theta$, with a non-negative loss function $L(z, \theta)$ on a sample $z = (\bar{x}, y) \in \mathcal{X} \times \mathcal{Y}$. Let $R(Z, \theta) = \frac{1}{n} \sum_{i=1}^{n} L(z_i, \theta)$ denote the empirical risk, given a set of data $Z = \{z_i\}_{i=1}^{n}$. We assume that the empirical risk is differentiable in $\theta$. The training data are supplied by a set of data holders.

### A.1  NON-IID SETTING

The main hurdle for Federated Learning is that not all data is IID. Heterogeneous data distributions are all but uncommon in the real world. To simulate a Non-IID distribution, we used Dirichlet distribution to split the training dataset as in related literature Hsu et al. (2019); Lin et al. (2020); Hoech et al. (2022); Yu et al. (2022). This distribution is parameterized by $\alpha$, which controls the concentration of different classes, as visualized in Figure 8. This work uses $\alpha \to 0.1$ for a non-IID distribution, as in related literature (e.g., Yu et al. (2022)).

### A.2  EXACT INFLUENCE

In simple terms, influence measures the marginal contribution of a data point on a model's accuracy. A positive influence value indicates that a data point improves model accuracy, and vice-versa. More specifically, let $Z = \{z_i\}_{i=1}^{n}$, $Z_{+j} = Z \cup z_j$ where $z_j \notin Z$, and let

$$\hat{R} = \min_{\theta} R(Z, \theta) \quad \text{and} \quad \hat{R}_{+j} = \min_{\theta} R(Z_{+j}, \theta)$$

where $\hat{R}$ and $\hat{R}_{+j}$ denote the minimum empirical risk of their respective set of data. The *influence* of datapoint $z_j$ on $Z$ is defined as:

$$\mathcal{I}(z_j, Z) \triangleq \hat{R} - \hat{R}_{+j} \tag{4}$$

Despite being highly informative, influence functions have not achieved widespread use in Federated Learning (or Machine Learning in general). This is mainly due to the computational cost. Equation 4 requires complete retraining of the model, which is time-consuming, and very costly; especially for state-of-the-art, large ML models. Moreover, specifically in our setting, we do not have direct access to the training data. In the following section, we will introduce a practical approximation of the influence, applicable in Federated Learning scenarios.

### A.3  INFLUENCE APPROXIMATION

The first-order Taylor approximation of influence, adopted by Koh & Liang (2017) (based on Cook & Weisberg (1982)), to understand the effects of training points on the predictions of a *centralized* ML model. To the best of our knowledge, this is the current state-of-the-art approach to utilizing the influence function in ML. Thus, it is worth taking the time to understand the challenges that arise

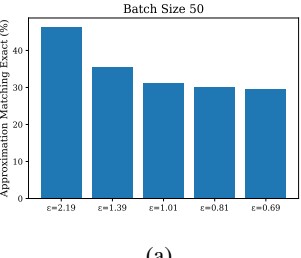 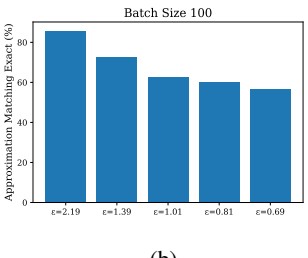 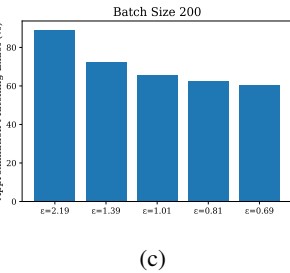

Figure 5: Comparison between different $\epsilon$. Comparing 5a to the other two figures, we can observe that there is a certain threshold of data that needs to be passed for *lazy influence* to be effective. After this threshold has been reached, adding more data only gives marginal improvement as can be seen by comparing 5b and 5c.

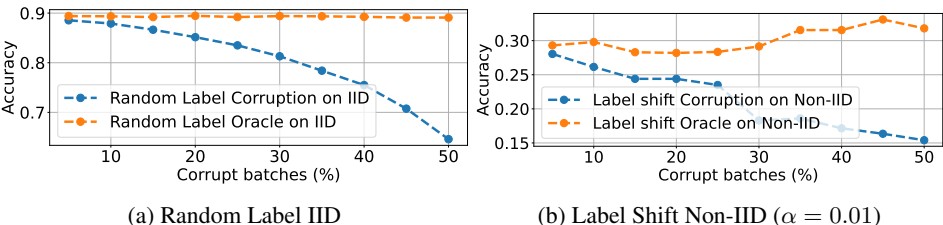

(a) Random Label IID   (b) Label Shift Non-IID ($\alpha = 0.01$)

Figure 6: Model accuracy relative to different mislabel rates (5% - 50%). These models have been trained over 25 communication rounds and 100 participants. We compare a centralized model with no filtering of mislabeled data (blue) to an FL model under perfect (oracle) filtering (orange). Note that the lower accuracy on the Non-IID setting is due to the fact that we are considering the most extreme non-IID case. This is where the majority of the participants have access to at most 1 class.

if we adopt this approximation in the Federated Learning setting. Which performs decently well compared to an exact influence evaluation as can be seen in Figure 5.

Let $\hat{\theta} = \arg\min_\theta R(Z, \theta)$ denote the empirical risk minimizer. The approximate influence of a training point $z_j$ on the validation point $z_{val}$ can be computed without having to re-train the model, according to the following equation:

$$\mathcal{I}_{appr}(z_j, z_{val}) \triangleq -\nabla_\theta L(z_{val}, \hat{\theta}) H_{\hat{\theta}}^{-1} \nabla_\theta L(z_j, \hat{\theta}) \tag{5}$$

where $H_{\hat{\theta}}^{-1}$ is the inverse Hessian computed on all the model's training data. The advantage of Equation 5 is that we can answer counterfactuals on the effects of up/down-scaling a training point, without having to re-train the model. One can potentially average over the validation points of a participant, and/or across the training points in a batch of a contributor, to get the total influence.

### A.4 CHALLENGES

Consider Figure 6 as a motivating example. In this scenario, we have participants with corrupted data. Even a very robust model (ViT) loses performance when corruption is involved. This can also be observed in the work of Li et al. (2021). Filtering those corrupted participants (orange line) restores the model's performance.

While Equation 5 can be an effective tool in understanding centralized machine learning systems, it is *ill-matched* for Federated Learning models, for several key reasons.

To begin with, evaluating Equation 5 requires *forming and inverting* the Hessian of the empirical risk. With $n$ training points and $\theta \in \mathbb{R}^m$, this requires $O(nm^2 + m^3)$ operations Koh & Liang (2017), which is *impractical* for modern-day deep neural networks with millions of parameters. To overcome these challenges, Koh & Liang (2017) used implicit Hessian-vector products (HVPs) to more efficiently approximate $\nabla_\theta L(z_{val}, \hat{\theta}) H_{\hat{\theta}}^{-1}$, which typically requires $O(p)$ Koh & Liang (2017).

|  | CIFAR10 | CIFAR100 |
|---|---|---|
| No. of Participants | 100 | 100 |
| Batch Size | 100 | 250 |
| Validation Size | 50 | 50 |
| Random Factor | 0.9 | 0.9 |
| Warm-up Size | 600 | 4000 |
| Final Evaluation Size | 2000 | 2000 |
| Load Best Model | False | False |
| Parameters to Change | 7690 | 76900 |
| Learning Rate | 0.001 | 0.001 |
| Train Epochs | 3 | 3 |
| Weight Decay | 0.01 | 0.01 |

Table 1: Table of hyper-parameters.

While this is a somewhat more efficient computation, it is *communication-intensive*, as it requires *transferring all of the (either training or validation) data* at each FL round. Most importantly, it *can not provide any privacy* to the users' data, an important, inherent requirement/constraint in FL.

Finally, to compute Equation 5, the loss function has to be strictly convex and twice differentiable (which is not always the case in modern ML applications). A proposed solution is to swap out non-differentiable components for smoothed approximations Koh & Liang (2017), but there is no quality guarantee of the influence calculated in this way.

## B  IMPLEMENTATION DETAILS

This section describes the base model used in our simulations and all hyper-parameters. Specifically, we used a Visual Image Transformer (VIT) Deng et al. (2009); Wu et al. (2020). The basis of our model represents a model pre-trained on ImageNet-21k at 224x224 resolution and then fine-tuned on ImageNet 2012 at 224x224 resolution. All hyper-parameters added or changed from the default VIT hyper-parameters are listed in Table 1 with their default values. The following hyper-parameters have been added to support our evaluation technique:

- **Random Factor**: this coefficient represents the amount of corrupted data inside a corrupted batch.

- **Final Evaluation Size**: an a priori separated batch of test data to evaluate model performance.

- **Parameters to Change**: number of parameters (and biases) in the last layer of the model.

Regarding reproducibility, we ran the provided (in the supplementary material) code for each dataset with seeds from the range of $0 - 7$.

### B.1  TERMINATION CONDITION

Different termination conditions have been used for our proposed solution and to retrain the exact influence. Our solution has only one termination condition, that is the number of local epochs $k$.

### B.2  COMPUTATIONAL RESOURCES

All simulations were run on two different systems:

1. Intel Xeon E5-2680 – 12 cores, 24 threads, 2.5 GHz – with 256 GB of RAM, Titan X GPU (Pascal)

2. Ryzen 5900X - 12 cores, 24 threads, 3.7 GHz - with 32 GB of RAM, RTX 2080 Super

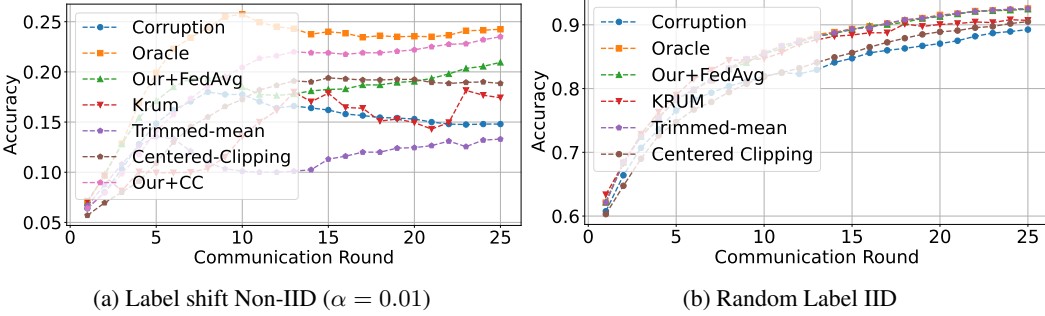

(a) Label shift Non-IID ($\alpha = 0.01$)          (b) Random Label IID

Figure 7: Model accuracy over 25 communication rounds with a 30% mislabel rate on CIFAR-10. We compare a centralized model with no filtering (blue) to an FL model under perfect (oracle) filtering (orange), KRUM (red), Trimmed-mean (purple), and our approach (green). Note that the jagged line for KRUM is because only a single gradient is selected instead of performing FedAvg.

|  | | Filtration Metrics | | |
|---|---|---|---|---|
|  | Distribution | Recall | Precision | Accuracy |
| CIFAR 10 | IID | 97.08 ± 3.51 % | 91.91 ± 7.15 % | 96.38 ± 2.83 % |
|  | Non-IID | 93.75 ± 5.12 % | 69.02 ± 6.28 % | 85.00 ± 3.28 % |
| CIFAR 100 | IID | 99.17 ± 2.20 % | 97.96 ± 2.30 % | 99.12 ± 1.27 % |
|  | Non-IID | 92.50 ± 5.71 % | 55.41 ± 3.94 % | 75.12 ± 3.76 % |

Table 2: Quality of filtration metrics for a setting with 100 participants, under strict worst-case privacy guarantees ($\varepsilon = 1$). Please see the supplement for the complete results.

## C   SOCIETAL IMPACT

Privacy advocacy movements have, in recent years, raised their voices about the potential abuse of these systems. Additionally, legal institutions have also recognized the importance of privacy, and have passed regulations in accordance, for example, the General Data Protection Regulation (GDPR). Our work provides practical privacy guarantees to protect all parties, with minimal compromise on performance. Furthermore, we allow data holders and collectors to be paid for their contribution in a joint model, instead of simply taking the data. Such incentives could potentially help speed up the growth of underdeveloped countries, and provide more high-quality data to researchers (as an example application, consider paying low-income farmers for gathering data in crop disease prevention Mohanty et al. (2016)).

## D   LIMITATIONS

The main limitation of our approach is that if the optimizer does not produce a good enough gradient, we cannot get a good approximation of the direction the model is headed for. The result of this is a lower score, and therefore a potentially inaccurate prediction.

Another potential limitation is the filtering of "good" data. These data may be correctly labeled, but including it does not essentially provide any benefit to the model, as can be shown by the accuracy scores in Figure 7. While this allows us to train models of equal performance with a fraction of the data, some participants may be filtered out, even though they contribute accurate data. This might deter users from participating in the future.

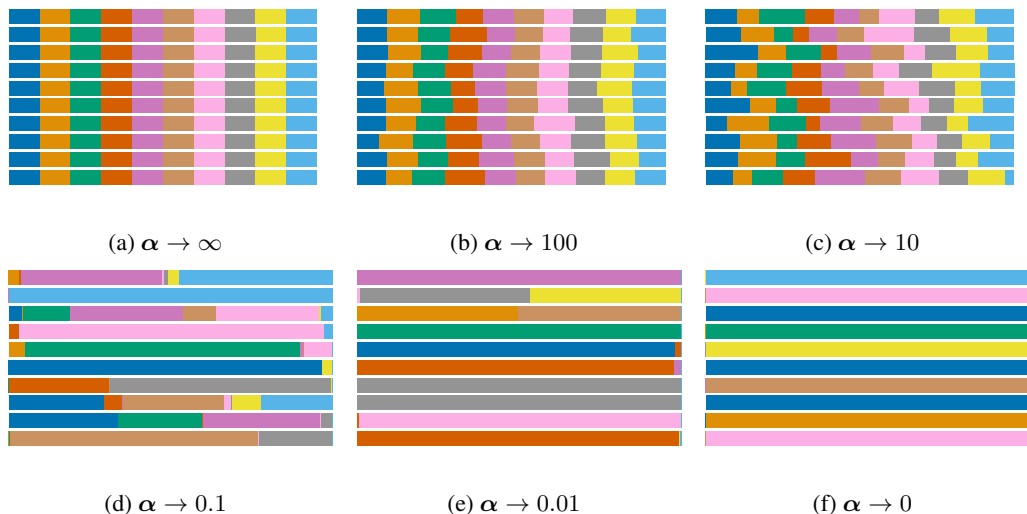

(a) $\alpha \to \infty$          (b) $\alpha \to 100$          (c) $\alpha \to 10$

(d) $\alpha \to 0.1$          (e) $\alpha \to 0.01$          (f) $\alpha \to 0$

Figure 8: Dirichlet distribution visualisation for 10 classes, parametrized by $\alpha$. $\alpha$ controls the concentration of different classes. Each row represents a participant, each color a different class, and each colored segment the amount of data the participant has from each class. For $\alpha \to \infty$, each participant has the same amount of data from each class (IID distribution). For $\alpha \to 0$, each participant only holds data from one class. In this work, we use $\alpha \to 0.1$ for a non-IID distribution.

# E   NUMERICAL RESULTS

We provide detailed results that include both the means and standard deviations. The metrics can be found in Table 2. The following subsections provide a more comprehensive analysis of the results, summarized in the main text due to space limitations.

We have conducted initial testing on CIFAR10 and CIFAR100 to explore the impact of various parameters on our model's performance. Our results, illustrated in Figures, 10, and 11 demonstrate the effect of both learning rate and number of epochs on filtration performance.

We observe a balance between recall and accuracy that varies based on the parameters. This balance can be seen in both the CIFAR10 and CIFAR100 datasets. Additionally, the best parameters for IID and Non-IID may differ. For instance, the best recall for Non-IID and IID is achieved with different parameter pairs, and CIFAR100 also has a distinct parameter pair for IID compared to Non-IID.

Finally, we examine the impact of various privacy guarantees ($\varepsilon$) and larger problem dimensions in Figure 9. Our findings show that a smaller federation is needed to achieve the same level of performance when data is IID, compared to when it is Non-IID.

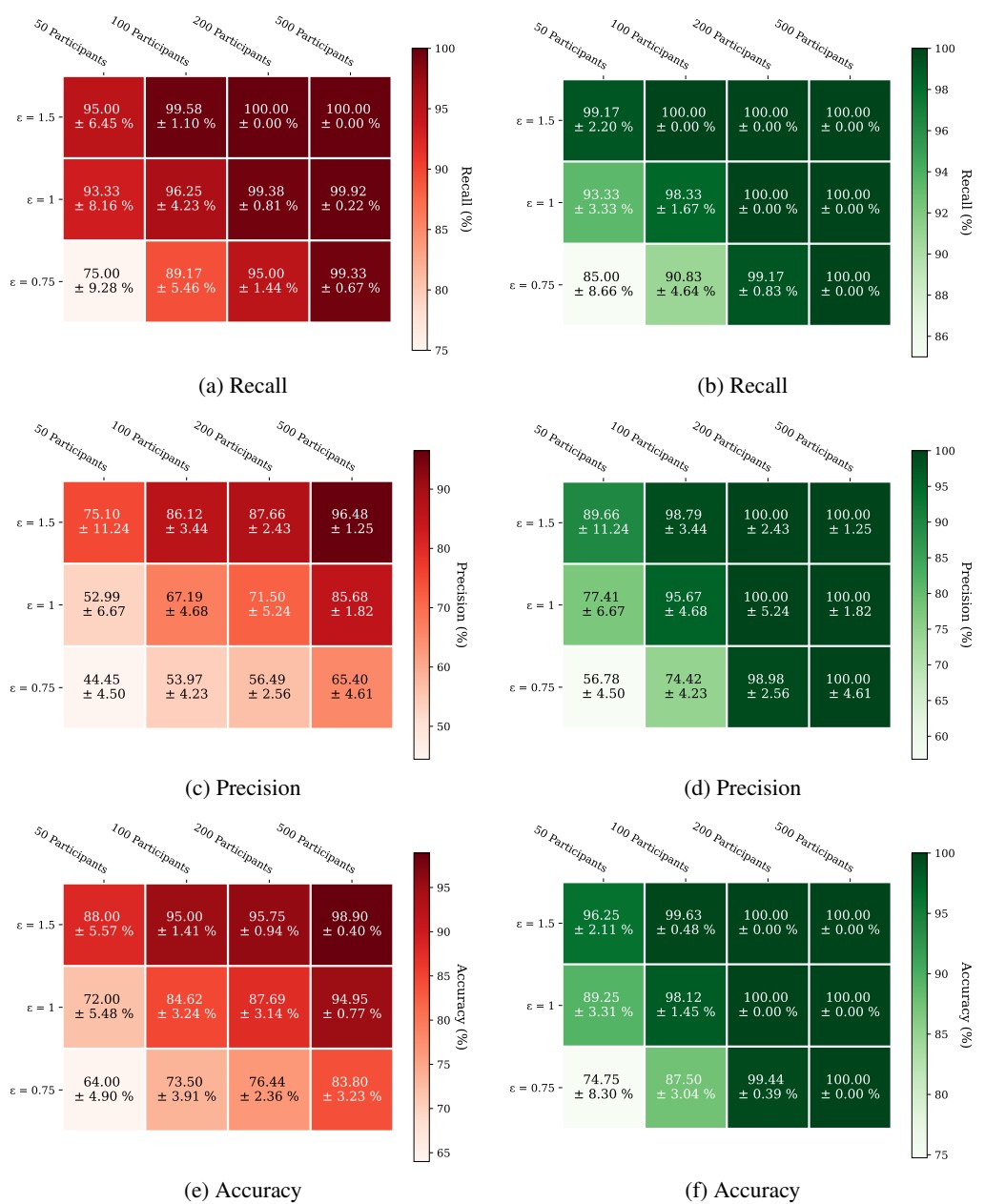

Figure 9: Recall (top), Precision (middle), and Accuracy(Bottom) on CIFAR 10, non-IID (left), IID (right), for increasing problem size (number of participants), and varying privacy guarantees ($\varepsilon$ – lower $\varepsilon$ provides stronger privacy).

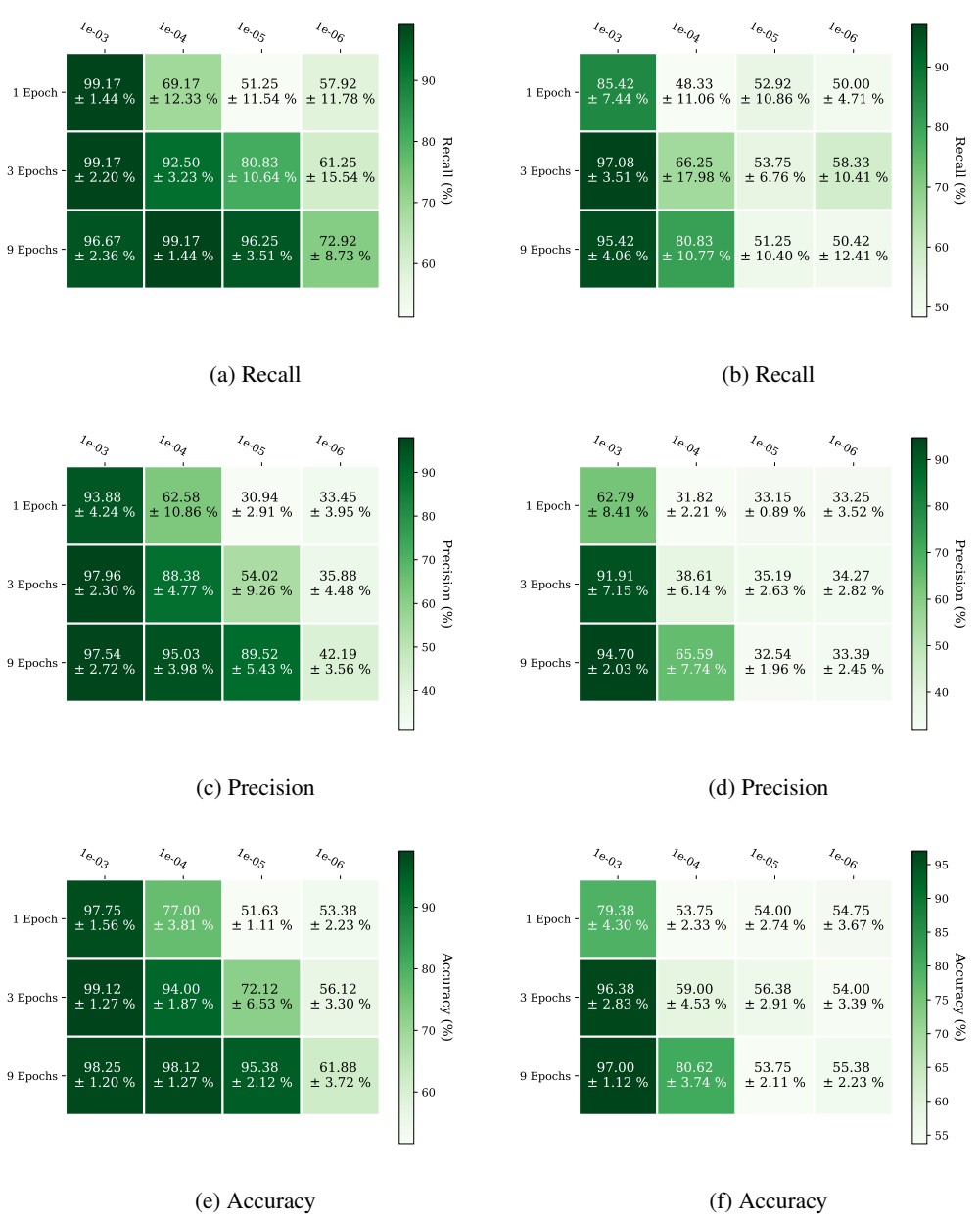

Figure 10: Recall (top), Precision (middle), and Accuracy(Bottom) on CIFAR 100 (left) and CIFAR 10 (right), IID, for different parameter pairs of learning rate and epoch count.

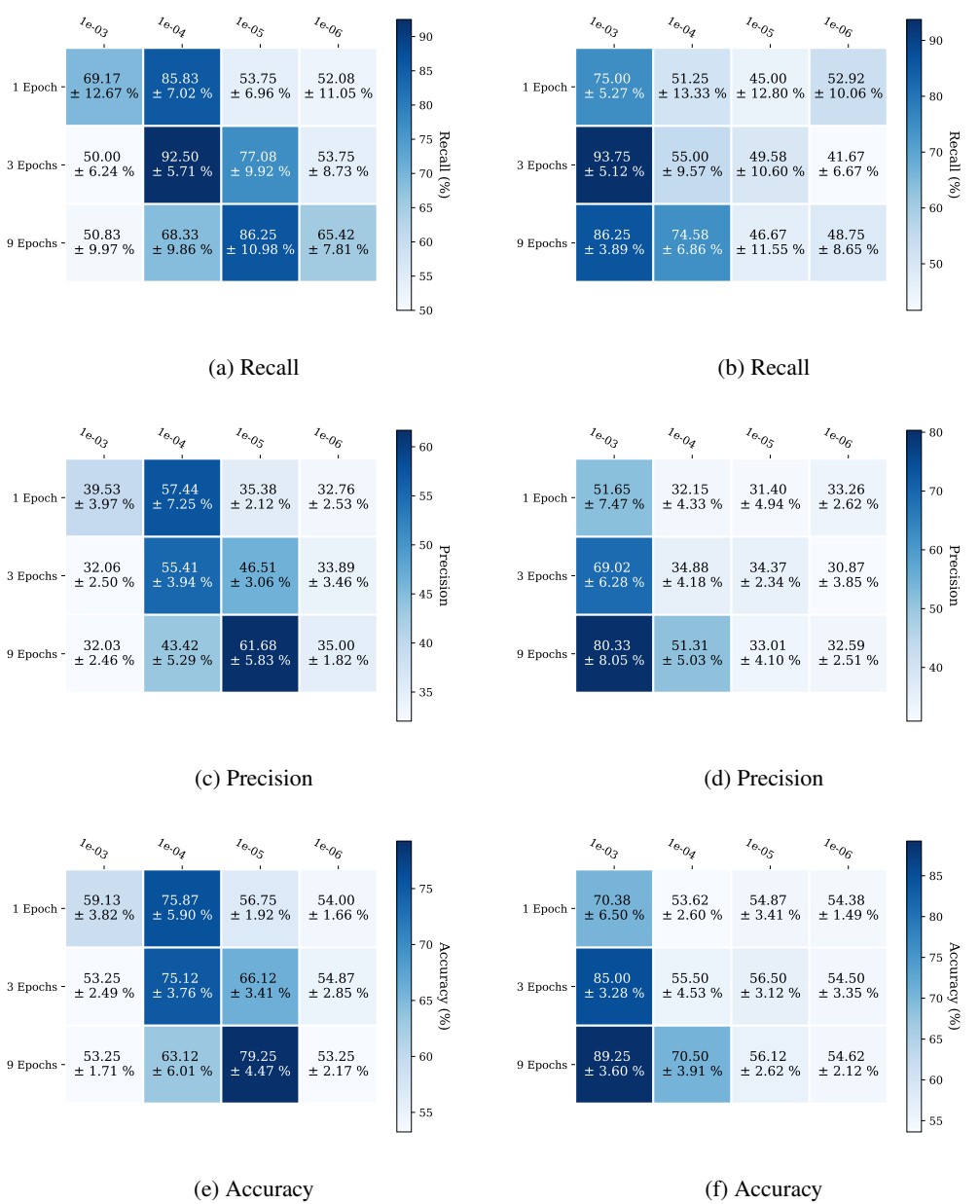

Figure 11: Recall (top), Precision (middle), and Accuracy(Bottom) on CIFAR 100 (left), and CIFAR 10 (right), non-IID, for different parameter pairs of learning rate and epoch count.

