# OpenReview forum: "Privacy-Preserving Data Quality Evaluation in Federated Learning Using Influence Approximation"
_ICLR.cc/2024/Conference — Submitted to ICLR 2024_

### Official Review · Reviewer_pxB7 · 2023-10-23

**Soundness:** 3 good
**Presentation:** 3 good
**Contribution:** 1 poor
**Rating:** 3
**Confidence:** 4

**Summary:**

This work presents a new method for data valuation in FL by using an approximated version of data influence (termed lazy influence) which can be used to perform data filtering and guide the training process

**Strengths:**

The work is well motivated and easy to follow. The framework that authors propose offers a novel solution to data selection (and performance improvement) problems in FL. Particularly nice to see that authors considered non-IID settings, as these can often really offset the contributions.

**Weaknesses:**

However, I have a number of concerns regarding the experimental setting, interpretation of DP and the scalability of the method.

One note on the epsilon value: it is incredibly misleading to say that the achieved score (<1) is lower (and hence better, or more private) than the rest of the literature. Epsilon does not exist in a vacuum, its meaning for the user is affected (among other things) by: modality, dataset composition, which part of FL falls under DP etc. That being said: it is possible to achieve the same MAGNITUDE of epsilon when these components are vastly different, but this also means that the interpretation of epsilon is no longer the same (i.e. saying that a randomised response epsilon of <1 is more private than DP-SGD of 10 is strenuous at best and fundamentally incorrect at most, as these cannot be directly compared against each other by magnitude alone).

Having an assumption of a holdout server-side dataset is a bit too strong and not necessarily standard for FL settings.

**Questions:**

By the description of the protocol I am not super convinced this is actually FL: it looks more like some variation of P2P learning, given that clients communicate with each other directly?

From what I understood this method is also not applicable to a general FL setting and forces the federation into synchronous FL? Otherwise I am struggling to see how the stale updates would be filtered+handled. Additionally, do all clients assume to be selected each round? The client selection section really does not expand on this.

Is CIFAR-based evaluation the only one considered? Not only are there no results in other modalities (which is typically a limitation, but not always a major one), but there are also no results on more complex image classification datasets either?

One other point which is missing from the discussion is the concept of influence altogether: here it is presented that positive influence improves the model and it is hence better to consider updates with positive influence. This is in general the case, but I have not seen any discussion on the meaning of influence i.e. higher influence does not always imply that model is improving (as it is possible, particularly in FL with small datasets, to overfit on individual clients). Same goes for tracking influence: having negative influence in round 1 does not prevent the client from having a positive one in a later one (and, in fact, does not always convey that certain clients are ‘better’, they may simply have simpler, more typical data). I would like authors to discuss/show what happens to influence of the same client over time w/out filtering to see if there is any long-term benefit.

Could authors also point me to how well the method performs (e.g. training time compared to vanilla FL) computationally? Since you mention how prohibitive Shapley values are, I would have expected how much better the proposed method performs.

Overall, I find this work to be rather limited in scope and to have several potential issues, which I would like to be clarified before I can recommend acceptance.

---

> ### Author Response · Authors · 2023-11-22
>
> We want to thank the reviewer for their comments. To answer the posed questions:
>
> 1) The proposed method is independent of the aggregation strategy and works with any type of Federated Learning.
>
> 2) The federation is not forced into synchronization. The main idea is to be able to assign a numeric value to a device's potential contribution to the federation. It can be used as a "right of passage" before a device is admitted into the federation. This process can be fully asynchronous. We leave the exploration of an asynchronous approach to future work, intuitively we believe the results would be similar.
>
> 3) We mostly focused on CIFAR-10 and CIFAR-100 (whose results can be found in the supplementary material), and we are planning to expand our work to different domains in the future.
>
> 4) Regarding the overfitting issue, it would be highly unlikely to happen, due to the fact that the different validation sets are sampled from different distributions. Intuitively, the long-term influence of a client would stay positive as long as their data is beneficial, though it would be an interesting direction to explore.
>
> 5) The proposed method, used as a tool for quality control, does not have a high communication and computation cost. In the case where we would run it every round scaling it would pose problems for larger federations. The worst-case communication cost would be O(nk), where k is a subset of devices used for evaluation.

---

### Official Review · Reviewer_KGyK · 2023-10-28

**Soundness:** 3 good
**Presentation:** 3 good
**Contribution:** 2 fair
**Rating:** 5
**Confidence:** 3

**Summary:**

This paper works on data filtering in Federated learning with differential privacy. The idea is to estimate the influence of each batch of training data by 1) first updating a small fraction of model with training data and 2) then evaluating the performance of the updated layer on validation data. During the process, suitable noise is added into the transmitted information so that differential privacy is enforced. Experiments show great improvement even in the non-IID settings.

**Strengths:**

The idea is elegant and easy to follow. Multiple strategies, like model freezing and bit compression are used to reduce communication. The experiment results look promising.

**Weaknesses:**

- Theoretical analysis, or some kind of high level intuition for the proposed method is highly appreciated. For instance, why can the proposed method work in non-IID settings? What type of data can be filtered out by the proposed method? Currently it is not clear to me why or when the proposed method can work.

- Data collection is very challenging for federated learning. In the proposed method, data is further split into training data and validation data. Will this hurt the performance of the model? For instance, if we just use all the data for training, will this be a much stronger baseline? In the current setting 1/3 data is used for validation. More justification is appreciated for this setup.

- Scalability seems to be a question. Say there are $n$ parties. Compared to standard federated algorithm like FedAvg, each party needs to conduct $n$ times more validation computation. And the overall communication seems to scale in $O(n^2)$ since essentially every pair of parties need to communicate with each other.

- Sign SGD [1] seems to be a related reference and needs to be cited.

[1] Bernstein, Jeremy, et al. "signSGD: Compressed optimisation for non-convex problems." International Conference on Machine Learning. PMLR, 2018.

**Questions:**

Is the overall distribution of training data and test data identical? If not, then it is not clear why validation data can be helpful for filtering training data.

---

> ### Author Response · Authors · 2023-11-21
>
> We want to thank the reviewer for their comments.
>
> To answer the posed weaknesses:
>
> 1) The high-level intuition behind why our proposed method works in a Non-IID setting is that based on the votes two clusters will form, one cluster with a lower average (the corrupted data) and the other with a higher average of votes. Empirically we have shown that this is the case.
>
> 2) The training and validation data don't have to be mutually exclusive. We just require some kind of baseline to evaluate model updates provided by other devices.
>
> 3) We solve the scalability issue by proposing a few solutions:
>     a) Take a subset of devices k reducing the complexity to O(nk),
>     b) Only run the algorithm to verify devices once when they want to join the federation,
>     c) Only run the algorithm when the model performance stops improving or degrades.
> 4) We would like to thank the reviewer for their suggestion
>
> To answer the posed question:
> 1) The overall distribution of the test and training data is only identical in the IID setting. When considering a Non-IID setting each device has its own training distribution, while the test data used to evaluate the global model is uniform from all classes.

---

### Official Review · Reviewer_633s · 2023-11-01

**Soundness:** 3 good
**Presentation:** 2 fair
**Contribution:** 3 good
**Rating:** 5
**Confidence:** 3

**Summary:**

This paper considers the problem of federated learning in the setting where we want to preserve privacy under local differential privacy, and want to be robust to mild corruptions. Specifically, corruptions in this setting are not the same as the adversarial corruption notion in other works in this area, such as Byzantine robustness. They assume the users and the servers are honest but curious, and a fraction of the data they have might be corrupted non-adversarially. Their approach draws insights from influence functions. They employ a method which they call lazy iteration that utilizes the influence signs of each new (local) update.

Their approach is as follows: at each round each contributor user privately sends their updated last layer to another set of users that act as validators. Then each validator checks whether this new update will improve accuracy over the data they have or not, and sends a private vote to the server. Privacy of the vote is both with respect to the data that the validator has and over the data that the contributor has used to provide the update. After that the server decides whether to accept the update of the contributor or not by checking whether the number of accept votes is above some threshold or not. They decide the threshold by doing $k$-means on the total number of positive votes and then setting the average of the two clusters as the threshold.

They run experiments for 25 communication rounds and compare their results with other methods in this area that provide byzantine robustness as their baseline, for example KRUM [Blanchard et al.] (2017), and Trimmed-mean [Yin et al.] (2018), and Centered-clipping [Karimreddy et al.] (2021). They run their experiments on CIFAR 10 with IID and random labeling and label shift and non iid data with label shift. The way they generate non-iid samples is by sampling from a Dirichlet distribution. They present their experiments and compare with the above Byzantine robustness baselines. Their methods can also be employed together with other Byzantine robust algorithms such as Centered Clipping.

**Strengths:**

In their experiments their performance is close to the performance of the oracle in the IID with random labeling setting and their approach, and their approach on top of centered clipping outperforms Byzantine robust methods in the label shift with non-iid data.

I think there's value in studying the setting where the verification of updates is decentralized. Most of the previous work focuses on the setting where the task of verification / filtering is done centrally.

Their approach can work on top of the Byzantine robust algorithms.

**Weaknesses:**

The main conceptual criticism I have is that it is not clear whether this method provides substantial improvements over a private version of Byzantine robust approaches or an alternative central model that is robust under mild corruption assumptions as in this paper. There's definitely some improvements in the experiments, but I think the cost of communication between verifiers and participants may outweigh that. For example, it seems like in this approach, in each round we are going to have quadratic in the number of users total communications, compared to the Byzantine robustness algorithms in previous work that only require a linear number of communications.

I have some other concerns that I will share in the questions section below.

**Questions:**

It is not clear whether the baselines in the experiments are also privatized or not.

In Algorithm 1, does each user send their updated last layer of the model to C as well?

It is mentioned that the output of the validation being private helps with the other users not being able to tailor their updates to the validation data. From adaptive data analysis we know that such an approach only holds out for a limited number of interactions. How many rounds of interactions / validation can this approach tolerate given say $n$ validation examples? I think this is important because the number of interactions with the hold-out validation data of each user seems to scale with the number of all users, which could be much larger than the amount of validation data a single user has.

It is mentioned that a drawback of other work int this area is the they may eliminate "minority" distributions due to their large distance relative do other model updates. Isn't that also the case in your setting as well? For example if a user has data that comes from a minority distribution, the rest of the users that have validation data that comes from a majority distribution would vote negatively for its update and therefore omitting that update.

It is mentioned that LDP is a generalization of DP, I'm not sure if generalization is the right word to describe the relationship here.

In the challenges section, it is mentioned that the Koh & Liang (2017) result requires, O(p) many operations but p is not defined anywhere.

In the paragraph after figure 2, it is mentioned that even under really strict privacy guarantees, the aggregated influence signs will match the true value in expectation. I'm not sure how to interpret this sentence. The way I interpret it is $\sum_i \mathbb{E}[v_i'] = \sum_i v_i$, which is not true, in fact we have $\sum_i \mathbb{E}[v_i'] = (1-p) \sum_i v_i$.

In Figure 3, it's not clear to me what communication rounds means here. Does it mean that for 25 rounds all users have sent their proposed updates to the server?

---

> ### Author Response · Authors · 2023-11-21
>
> We want to thank the reviewer for their comments.
>
> To address the main concern: our proposed method can be used indifferently to the aggregation strategy of the federation. It can be used to access the data quality of devices before they join a federation and therefore guarantee good quality data.
>
> To answer the posed questions:
>
> 1) The baselines are not privatized. The only privatization applied in this paper is for the lazy influence evaluation scheme.
>
> 2) No, our proposed method is completely indifferent to the aggregation strategy of the federation.
>
> 3) This is a good question, if we were to run our proposed method every iteration this would become a problem. We have not analyzed this scenario yet and would like to thank you for pointing this out.
>
> 4) This is correct if the device has no other devices that share any similarity in distributions. Given a more realistic scenario where there is a small minority instead of a single device, the algorithm can potentially differentiate them from uninformative data. However, more experiments need to be conducted in this scenario to say for certain.
>
> 5) You are completely right, we will correct the wording.
>
> 6) We apologize for forgetting to include what p is, it represents the number of parameters of the model.
>
> 7) To clarify, we meant with that statement that the sign of the aggregated influence signs will match the sign of true value in expectation.
>
> 8.) The communication rounds are standard FL communication rounds, which have been performed with a subset of agents. This subset of agents consists of agents who have successfully passed the filtration.

---

> > ### Comment · Reviewer_633s · 2023-11-22
> >
> > I thank the authors for their response and answering my questions.
> >
> > Main concern: I understand that this model can be used on top of any aggregation strategy. However it's not clear to me how to compare the improvement to the setting where we're not using this method and are only using a privatized Byzantine robust style algorithm. The accuracy in the setting where we use the technique in this paper on top of the existing algorithms may be better, per round of standard FL. The issue is that the cost of the communication per FL round is going to be quadratic in terms of the number of users when the algorithm in this paper is used, while the cost of communication scales linearly in terms of number of users when for example a privatized Byzantine robust style algorithm is used. So would the right comparison to understand the performance of this algorithm be comparing $1$ FL round of the algorithm in this paper vs. $1$ FL round in a Byzantine robust style algorithm, or would it be $1$ FL round of the algorithm in this paper vs. $n$ rounds of FL in a Byzantine robust style algorithm, where $n$ is the number of users? It's not clear.
> >
> > I hope I have made my concern clear.

---

### Official Review · Reviewer_eJdW · 2023-11-01

**Soundness:** 2 fair
**Presentation:** 3 good
**Contribution:** 3 good
**Rating:** 5
**Confidence:** 3

**Summary:**

The paper proposes a simple yet effective approach that utilizes a new influence approximation called ”lazy influence” to filter and score data while preserving privacy. To do this, each participant uses their data to estimate the influence of another participant’s batch and sends a differentially private obfuscated score to the FL server.

**Strengths:**

- The approximation of the influence is efficient, which reduces the computational complexity for the influence score.
- The proposed method increases the model's performance on the benchmark datasets.
- Leveraging strong and well-known privacy-preserving mechanism.

**Weaknesses:**

- The work seems incremental with limited novelty since it applies existing works for privacy protection.
- Lacking of theoretical analysis for privacy protection.
- The computation over-head is high since at every epoch, a client has to communicate with all other clients and the FL server.

**Questions:**

1. How the privacy accumulation over multiple updating rounds is computed in your proposed method?
2. McMahan et al. 2017 proposed User-level DP and RAPPOR provides local differential privacy. Therefore, in the proposed method, what is the level of privacy that you are providing?
3. Since using the proposed method from McMahan et al. 2017, the gradients are clipped which will clip out the information induce from local data. Therefore, the data filtering process might filter out important data points. What is the impact of the clipping bound toward the data filtering process ?

---

> ### Author Response · Authors · 2023-11-21
>
> We want to thank the reviewer for their comments. To answer the posed questions:
>
> 1) We apologize for not making it clear in the submission, but our proposed method is meant as a tool to detect and filter out devices that do not contribute to the improvement of the model. All our experiments were run with the method filtering out devices at the start and preventing them from participating. The following rounds have then been standard FL rounds.
>
> 2) We provide 2 levels of privacy:
>       a) User-level DP for the transmitted gradient update which is used to evaluate the device quality.
>       b) RAPPOR to make the client votes differentially private.
> 3) The effect of the clipping in our proposed method is the loss of quality evaluation accuracy. To combat this we require a larger amount of devices to maintain the same level of filtering accuracy, as can be seen in Figure 4.

---

### Comment · Area_Chair_QCkh · 2023-12-04
**Question about privacy guarantees**

Dear authors,

I have a couple quick questions regarding the differential privacy guarantees of the protocol. Can you please help clarify them?

1. It is claimed that the last layer of $\tilde{\theta}_t$ that is sent to the validators is protected with local DP (at user-level, as indicated by reply to reviewer eJdW). However, I struggle to understand how this can then be useful for validation. Namely, by the definition of local DP, this means that the probability that $A$ is filtered and the probability that $A$ is not filtered must be very close (for small $\epsilon, \delta$) regardless of the operations we do on top of this noisy last layer. So this should not give anything much better than a random coin toss. Can you please clarify what my misunderstanding is?

One possible discrepancy here is that the noises added to the last layer are *independent* for different validators. However, in this case, composition of DP has to be applied; otherwise, we would not get protection against collision of validators.

2. Related to the question above, if the last layer is protected with DP noise, then what is the $\delta$ used here? And how is the budget split between this and the RAPPOR for voting?

---

### Meta-Review · Area_Chair_QCkh · 2023-12-06

**Metareview:**

This paper provides a new approach to data filtering in the federated learning (FL) setting using a new notation called *lazy influence*. Recall that *influence* roughly measures the difference in accuracy of the model if we train with and without a particular datapoint. This serves as a good proxy for whether the datapoint is useful (or should be filtered out). However, computing influence is expensive because we have to retrain the model. In this paper, the authors propose to *lazy influence* where we only measure the accuracy change with respect to a small number of gradient descent steps. This dramatically reduces the computational cost. To use this for filtering in FL, each client (acting as "data contributor") would run a small number of gradient descent steps using their own data and then send their new model to other clients (acting as "validators") who computes the accuracy. These accuracies are aggregated in FL manner and the examples are filtered out if the accuracy change is below a certain threshold. Noises are also added during these processes to achieve differential privacy (DP). Finally, to reduce the communication, the authors also propose freezing most of the model during lazy influence evaluation; this means that each client only needs to send the non-frozen part for validation purposes. The experiments show good performances of this approach even when the data is highly corrupted and non-iid.

### Strengths

- This paper introduces a concept of *lazy influence* which is (much) less computationally intensive to compute than influence but can still be used for the purpose of data filtering. It seems plausible that this concept will be useful beyond the context of this paper.

### Weaknesses

- Differential privacy claims lack details and rigorous proofs. This makes it very hard to understand the exact protection the protocol provides in terms of privacy. This is especially concerning given that the partially trained model of each contributor client is sent directly to validator client (whereas, in typical FL, at least some aggregation is performed).

- Since the (noisy) last layer of each contributor has to be sent to every validator, it seems that the communication here will be quadratic (at least in the batch size), which might effect the scalability of the protocols. Although this seems to be mitigated somewhat by the fact that only a small part of the model is sent, the issue is not properly discussed in the paper.

- Lack of discussions on potential downsides of influence-based filtering, and whether these carry over to lazy influences.

- The empirical evaluation is limited; it involves just a single round of FL and small number of clients.

**Justification For Why Not Higher Score:**

The main issue here lies with the privacy claims, which are not rigorous. (In fact I'm fairly confident that their DP claims are incorrect altogether. See my questions to authors below.) Due to this, I don't think the paper can be accepted until the privacy models and claims are made completely clear.

**Justification For Why Not Lower Score:**

N/A

---

### Decision · Program_Chairs · 2024-01-16

Reject